biomechanics, biophysics, biomaterials

gecko locomotion, adjustable distributed control, animal–environment interaction, adaptability, manoeuvrability

**Authors for correspondence:**
Zhendong Dai
e-mail: zddai@nuaa.edu.cn
Robert J. Full
e-mail: rjfull@berkeley.edu

# Role of multiple, adjustable toes in distributed control shown by sideways wall-running in geckos

Yi Song[1,2], Zhendong Dai[1], Zhouyi Wang[1] and Robert J. Full[2]

[1]College of Mechanical and Electrical Engineering, Nanjing University of Aeronautics and Astronautics, 29 Yudao Street, Nanjing 210016, People's Republic of China
[2]Department of Integrative Biology, University of California, Berkeley, CA 94720, USA

YS, 0000-0001-9239-5377; ZD, 0000-0002-1276-7466; ZW, 0000-0002-4842-3470;
RJF, 0000-0001-8435-5279

Remarkable progress has been made characterizing one of nature's most integrated, hierarchical structures—the fibrillar adhesive system of geckos. Nonetheless, we lack an understanding of how multiple toes coordinate to facilitate geckos' acrobatic locomotion. Here, we tested the control function of gecko toes by running them on vertical substrates varying in orientation, friction and roughness. Sideways wall-running geckos realigned the toes of their top feet upward to resist gravity. Toe contact area was not compromised, but redistributed. Geckos aligned all toes upward to resist slipping when encountering low-friction patches during sideways wall-running. Negotiation of intermittent slippery strips showed an increased contribution of particular toes to compensate for toes that lost adhesion. Increasing substrate roughness using discrete rods perpendicular to sideways locomotion resulted in geckos bending and/or rotating toes to conform to and even grasp the rods, with potential forces more than five times body weight. Geckos increase their effectiveness of manoeuvrability in demanding environments by taking advantage of the distributed control afforded by multiple toes. Our findings provide insight on biological attachment and offer inspiration to advance gecko-inspired robotics and other biomimetic applications.

## 1. Introduction

Gecko toes possess the most hierarchically integrated natural structures spanning seven orders of magnitude in size [1], demonstrating a creative evolutionary solution to a functional problem [2]. Reviews [3–6] detail the extraordinary progress of defining the hierarchy where emergent adhesive properties arise from intermolecular forces that depend on nanoscale interactions of billions of spatulae on millions of setae arranged in fields attached to a series of leaf-like scansors connected to tendons found in compliant toes. Adhesion via multiple potentially adjustable toes appears to facilitate the extraordinary manoeuvrability of geckos [7] on diverse terrain [8–12]. However, our understanding of how multiple toes on a foot coordinate to permit effective engagement of this integrated hierarchical structure is lacking. A major challenge is the determination of how toe adhesive control rapidly modulates the foot forces that facilitate reliable attachment and fast locomotion, especially on natural substrates [6].

To connect toe function and load-bearing in feet, Russell & Oetelaar [13] studied the static clinging of geckos on a vertical wall in different positions: head-up, head-down and sideways. They predicted that the third toe representing the anatomical midline of the foot might most closely align with the gravity vector. They also hypothesized that more toes might be aligned with the gravity vector independent of orientation. Instead of these general trends, each body

orientation had its own pattern of digit positions and clustering [13]. A variety of toe span patterns provide sufficient levels of support regardless of body orientation on vertical surfaces. Static station keeping requires only a small subset of the toe capacity to support body mass [13,14]. Toe function is probably more related to dynamic climbing during which patchy contact frequently occurs.

Whole-foot forces in geckos have been measured during level running [15], vertical climbing [9], sideways wall-running [16] and inverted locomotion [17]. Each orientation reveals differential foot function because foot forces vary in both magnitude and direction. Yet how multiple toes coordinate to deliver the reaction forces necessary for locomotion remains unknown. During downhill locomotion, geckos reverse their hind feet allowing the directional toe adhesive system to be used as a brake and a stabilizer [18]. In gecko-inspired robots, Kim *et al.* [19] emphasized the need to include toes in the hierarchical compliance, along with the ankles, legs and body. Robots with multiple toes have shown higher attachment reliability [20]. Multi-level conformability and redundancy appear to be critical, especially on variable natural surfaces.

Natural substrates can be rough, undulate and unpredictable, with only patchy areas for contact [10]. Nanorough surfaces have been shown to affect the probability of seta [21] and setal field [10] attachment. Micro-roughness can challenge lamellar adhesion on surfaces with amplitudes and wavelengths similar to the lamella length and inter-lamella distance [22]. Geckos climbing diverse substrates show the greatest whole-body acceleration on the smoothest surface, probably due to less toe slipping [8]. A recent symposium on gecko adhesion called for a greatly expanded effort to begin to quantify the rock and plant micro-topography exploited by geckos to define patches available for adhesion, as well as perturbations [23,24].

Here, we test four hypotheses of the role of multiple, compliant toes of geckos by estimating toe orientation using high-speed videos, toe contact area via frustrated total internal reflection (FTIR) [25], and ground reaction force using three-dimensional sensors. We focus primarily on sideways wall-running for it appears most challenging for toes because gravity is more decoupled from forward motion.

First, we hypothesize that *toes will share the load during locomotion*. Although geckos can support their bodies with a single toe that points upward, we propose that toe adhesion will vary in magnitude and direction when generating ground reaction forces during a step (figure 1b).

Second, we hypothesize that *toes will realign when the load is altered due to gravity*. We will test this hypothesis by comparing sideways wall-running to vertical climbing (figure 1a,b). We propose that toes will alter their orientation and contact area to sufficiently adhere in the upward direction, while still maintaining the fore–aft forces for forward locomotion [16].

Third, we hypothesize that *toes will adjust to resist slip perturbations*. We will test this hypothesis during sideways wall-running by placing slippery patches and strips in the geckos' path (figure 1e2,e3). For large slippery patches, we propose that toes will resist sliding by rapidly aligning against gravity. For low-friction distributed strips, we predict that toes remaining in contact will adjust their orientation and contact area to compensate for neighbouring toes that have lost contact.

Fourth, we hypothesize that *compliant toes will adjust to rough terrain*. Here, we will place a series of vertical acrylic rods with diameters comparable to a gecko's toe length in the path of sideways wall-running geckos (figure 1a,e4). We propose that gecko toes will conform to the rough, area-reduced terrain, and perhaps grasp the perturbations, thereby producing adequate forces to balance gravity and locomote.

# 2. Materials and methods

## (a) Animals
We used 14 tokay geckos (*Gekko gecko*) in total. Seven of them ($95.5 \pm 22.9$ g) were involved in the experiments at NUAA, China, and the other seven ($74.8 \pm 10.4$ g) were studied at UCB, USA. Tests done at NUAA were approved by Jiangsu Association for Laboratory Animal Science and the Jiangsu Forestry Department, and those conducted at UCB were approved by the Animal Care and Use Committee as mandated by the U.S. Animal Welfare Act and Public Health Service Policy. No animals were injured in any experiments.

## (b) Experimental methods
The measurement of reaction force and contact area (i) was conducted at NUAA, whereas all other experiments (ii–v) were done at UCB.

### (i) Measurement of reaction force and contact area
We designed a vertical track consisting of FTIR-enhanced acrylic sheets ($150$ mm $\times 35$ mm $\times 3$ mm) and three-dimensional force sensors [16] (figure 1b). When geckos climbed in the aisle upward, we collected the contact images highlighted by the FTIR and the reaction force at their feet using synchronized high-speed cameras and NI DAQ model (https://doi.org/10.6078/D1ZD6C [26], movie S1, methods). Given that the contribution of left and right feet of vertically climbing geckos are equal [9,16], we collected and analysed the force and contact area of contralateral feet (top left and bottom right, figure 1d) from seven individuals.

### (ii) Running in orthogonal directions on wall
Using the FTIR to highlight contact regions, we built another Plexiglas wall with a track that allowed sideways running (figure 1a,c; [26], movie S1). We measured the orientation and contact area of toes using a high-speed camera while geckos ran along the track sideways. We rotated the track to be upward and measured the orientation and contact area of toes of the same individuals in upward climbing as a control (see [26], methods).

### (iii) Sideways running on slippery surfaces
A slippery patch (figure 1e2; Teflon, $80$ mm $\times 130$ mm $\times 0.1$ mm) and a sequence of slippery vertical strips (figure 1e3, Teflon) with width ($w$) of 5 mm and gaps ($D$) of 10 mm were pasted on the sideways orientated vertical track, respectively (figure 1a; [26], movie S2). When geckos ran over such adhesion-resistant surfaces, we measured the alignment of toes through a high-speed camera ([26], methods) with the sideways running of the same individuals on the non-slip track as their own control.

### (iv) Sideways running on area-reduced rough substrates
We reduced the available area of substrate by replacing the middle of the above track with vertically aligned acrylic rods (figure 1e4). The diameter ($d$) of rods ranged from 6.4 to 12.7 mm, whereas the distance ($D$) between rods

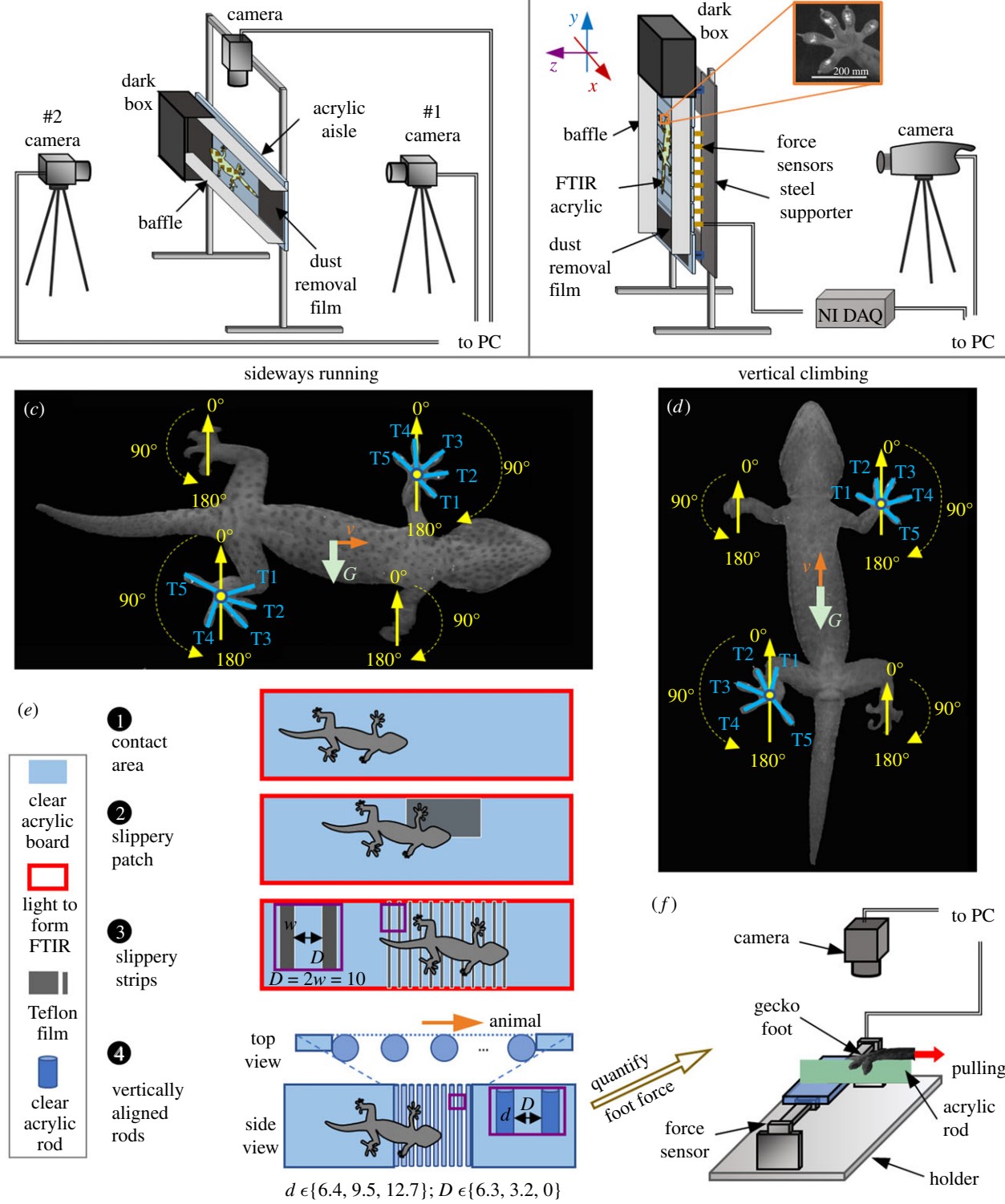

**Figure 1.** Experimental approach. (*a*) Apparatus to study toe deployments on sideways, slippery and/or area-reduced substrates. (*b*) Apparatus to measure vertical running reaction forces and contact areas. The force was measured in the coordinate system shown in the figure relative to a foot. Definitions of the orientation of toes in sideways (*c*) and upward (*d*) climbing. The letters '*G*' and '*v*' and corresponding arrows indicate the directions of gravity and motion, respectively. (*e*) Detailed design of sideways FTIR (*e*1), slippery patch (*e*2), slippery strips (*e*3) and area-reduced rough substrates (*e*4). '*d*', '*w*', '*D*' in mm. (*f*) Set-up to measure the grasping force of geckos' feet on rods that were used to construct the area-reduced substrates.

correspondingly decreased from 6.3 mm to 0 mm. We filmed the toes of the top feet of geckos with synchronized high-speed cameras as they ran sideways on the modified track ([26], movie S3).

### (v) Estimation of attachment on rods
We constructed an apparatus to measure the shear force of feet of same individuals across and along acrylic rods

(figure 1*f*; [26], movie S4) whose diameters ranged from 6.4 to 38 mm. The experiment was conducted following the methods used by Gillies *et al.* [22] ([26], methods). A flat acrylic was used as a control substrate. The ratio of the maximum force of a foot on rods over its maximum force on the control surface was calculated to represent its attachment capability.

## (c) Calculation of resultant foot contact

To best define the directionality of toes, we represented each toe by a vector showing orientation ($\theta_i$) and magnitude as the effective contact area ($A_i$). We calculated the resultant foot contact area in lateral ($A_x$) and upward ($A_y$) directions using the equation

$$A_x = \sum_{i=1}^{5} A_i \sin\theta_i, \quad A_y = \sum_{i=1}^{5} A_i \cos\theta_i. \tag{2.1}$$

## (d) Statistics

We conducted statistical analyses with SPSS19 (IBM Inc., NY, USA; [26], methods). We applied general univariate linear model (GLM) analysis and linear regression analysis (LR) to determine the relationship between foot force and resultant foot contact area. Repeated-measures ANOVA tests with Scheffe method for post hoc contrast analyses were applied to other comparisons. A significance level of 0.05 was used for all statistical tests. The statistical results are listed in the [26], table S4–S7.

# 3. Results and discussion

## (a) Toes vary in contact area, force and orientation during upward climbing

We measured the contact area and orientation of toes and reaction forces of the feet of geckos when they trotted upward ([26], figure S1a) by using FTIR-enhanced acrylics connected to three-dimensional force sensors (figure 1b,d). All toes varied in the magnitude of contact area and direction (figure 2b,e) as they generated foot reaction forces (figure 2a, d) during a step. Assuming the unidirectional adhesive toes of geckos can be represented by vectors, we calculated the contact at each foot in lateral and upward directions through equation 1. The shear forces (i.e. Fx & Fy, the solid lines in figure 2c,f) in both directions for all feet shared similar trends with the corresponding resultant contact area (i.e. Ax and Ay, the dashed lines in figure 2c,f) during the whole stance phase, yielding linear relationships between the shear force and resultant contact (figure 2g–i; LR, $p < 0.001$; see F-values and degrees of freedom in [26], table S4). There were no significant differences between the linearities in x and y directions (figure 2g–i; GLM, $p \geq 0.68$) or at different stance phases ($p = 0.06$). We found an average shear stress of 166.1 mN mm$^{-2}$ (adjusted $R^2 = 0.91$, $p < 0.001$; the 95.0% confidence interval was 162.2–170.0), comparable to results from previous studies [14].

Although geckos can support their bodies with single toes that point upward, they instead shared the load with distributed toes during climbing. Although load-sharing has not been found within a toe [25], the prevention of load concentrations could be significant within a foot [27] and among feet [9,17]. Feet divided into toes offer the opportunity for enhanced distributed control by adding more placement options to the hierarchy, reducing the possibility of load concentration. Furthermore, forming Y-shaped configurations with opposing toes can enhance the stability of attachment [28]. Compared with undivided feet, multiple toes differing in attachment force and orientation represent opportunities to distribute control by increasing the probability of establishing a secure foothold, especially during dynamic locomotion on challenging terrain [29]. The dependency between shear force and contact area lays the foundation of geckos regulating foot force by controlling distributed toes.

## (b) Toes realign when load is altered due to gravity

While running sideways along an acrylic wall (figure 1a,e1), geckos also used trotting gaits ([26], figure S1b and movie S1). Although the direction of gravity was changed by 90° relative to their trunks, their velocities were as fast as in upward climbing (sideways 0.90 ± 0.16 m s$^{-1}$, upward 0.84 ± 0.16 m s$^{-1}$). Successful wall-running requires the force generated at diagonal touching feet (e.g. top front + bottom hind) balance gravity. With FTIR highlighting and high-speed cameras recording contact (figure 1c), we compared the orientation and contact area of toes at the mid-stance phase in sideways running with upward climbing of the same individuals as a control (figure 3a,b; [26], results; see [26], table S5 for statistics).

Relative to upward climbing, sideways running geckos realigned their toes to sufficiently adhere in the upward direction (figure 3c with upward climbing rotated 90° clockwise; [26], movie S1), while still maintaining the fore-aft forces for forward locomotion [16]. Sideways wall-running geckos actively rotated all toes at top front feet upward by 12–20° ($p < 0.001$) and significantly increased their contact area by at least 42% ($p < 0.001$), thus providing a greater shear force in upward direction (figure 3c2). For the top hind feet during sideways running, geckos significantly reduced contact areas of the first two toes ($p < 0.001$), enlarged those of the last three toes ($p < 0.001$) and realigned all toes upward (figure 3c1; $p \leq 0.01$), shifting the direction of shear force by more than 90°. The toes at bottom front feet were also realigned (figure 3c4), with the first toes contributing major force in the upward direction ($p < 0.001$), while other toes barely showed contact. Limited by their skeletons, joint configurations and muscles of the hind feet [30], geckos might be less able to shift their toes at bottom hind feet to an upward orientation. As an alternative solution, the toes pointing downward reduced contact area, while the other toes shifted orientation and contact ($p < 0.001$) with the first toes to form a Y configuration with the fifth toes of the bottom hind feet (figure 3c3). In addition to the adjustment of orientation and contact area of toes, we noticed that adhesion frequencies of toes were also changed at all feet (figure 3a,b).

The realignment of toes with changing orientation has been shown in some geckos that cling to walls statically [13], but not all [31]. Geckos descending inclines can rotate their hind limbs opposite to the travel direction [18]. Although the orientations of toes we observed in running geckos showed similarities to those of static wall clinging in Bibron's geckos [13], the contribution of each toe was not consistent with predictions. It was suggested that toes in the upper quadrants must play significant roles in counteracting gravity by passive adhesion [13]. However, particular toes, rather than all toes that pointed upward, dominated foot attachment in both upward and sideways wall-running (figure 3a,b). The first and second toes at the top hind feet of sideways wall-running geckos often showed neglectable contact (figure 3c1). When the motion direction changed, the dominant toes also changed (figure 3c). Since there were no significant shifts in the orientation of toes during the stance phase ([26], movie S1), we conjectured that

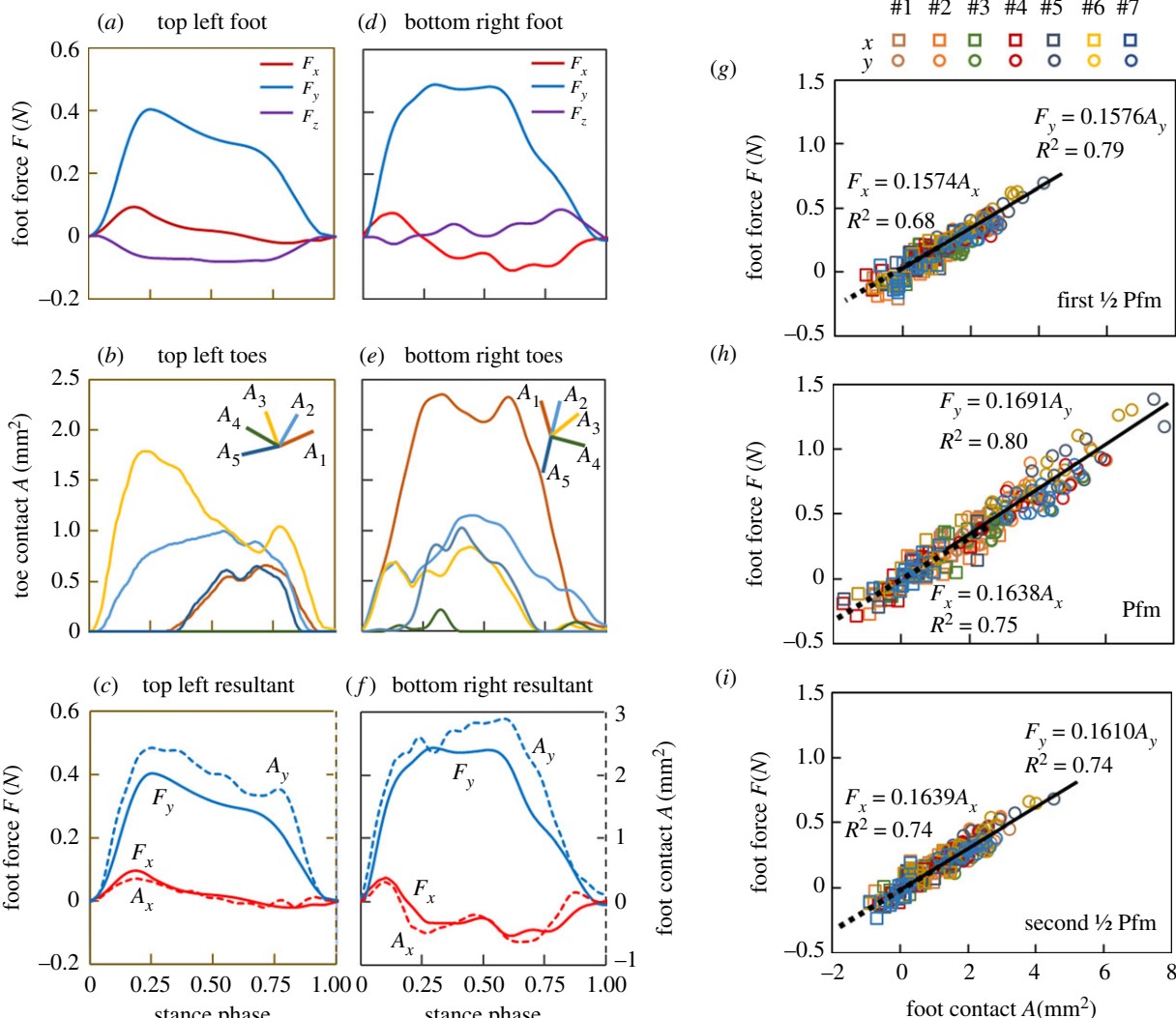

**Figure 2.** The contact area and reaction force obtained from upward climbing geckos. (*a,b,d,e*) Typical reaction force and contact area obtained at a top left foot (*a,b*) and a bottom right foot (*d,e*). (*c,f*) show the resultant contact (dashed lines) and shear force (solid lines). For the top feet, the angles between measured shear and normal forces were 11.2 ± 4.6°. (*g–i*) show the relationship between the resultant contact area and shear force at feet when the force increased to their half of peak force (first ½ Pfm, *g*), peak force (Pfm, *h*) and decreased to their half peak (second ½ Pfm, *i*), respectively. The squares and circles indicate the values in *x* and *y* directions, respectively, whereas the colours represent individuals. Two linear regressions (*x* and *y* directions) are shown for each phase. Positive lateral wall-reaction forces ($F_x$) correspond to forces where feet pull inward toward the trunk. The statistics can be found in [26], table S4.

geckos could actively configure toes to produce shear force during locomotion. During upward climbing, the top and bottom feet showed equal resultant contact areas in the upward direction (top 3.86 ± 1.38 mm², bottom 3.78 ± 1.72 mm²; $p = 0.54$), whereas during sideways running, geckos relied on the top feet for 80% of the vertical shear against gravity (figure 3*d*) [16]. Surprisingly, the resultant vertical contact areas of contralateral feet of sideways running geckos were not reduced below that seen in upward climbing, but were actually even larger (figure 3*d*). This result illustrates that the distributed control among toes could be achieved by adjusting the orientation and force of toes individually or jointly, without compromising the resultant contact performance.

## (c) Toes adjust to resist slip perturbations
Animals must cope with natural terrains that are not ideal for attachment. Insects can use claws, spines and adhesive pads to counter foot slipping [32], and even rely on the synergistic function of distributed claws and adhesive pads at a toe to enhance their attachment if the force generated at the sub-structures is insufficient [33]. For geckos, the setae do not necessarily exhibit decreased adhesion or friction characteristic of slipping, representing the transition from static to kinetic contact mechanics. Instead, friction and adhesion forces could increase at the onset of sliding and can continue to increase with shear speed [34]. We added adhesion-resistant patches and strips to the track (figure 1*a,e*2,*e*3) when geckos ran sideways. We found that geckos rely on distributed toes to enhance the attachment of corresponding feet during slipping perturbations.

## (i) Slippery patch perturbation
If the top feet of vertically climbing geckos fail to adhere, a tail reflex can maintain trunk position and resist over-turning [35]. Here, when all toes on a foot were inoperative, sideways wall-running geckos resisted sliding caused by gravity (figure 1*a,e*2) with highly directionally aligned digits

**Figure 3.** Comparison of toe deployment of geckos running in orthogonal directions. (*a*,*b*) The orientation and contact area of toes on all feet at mid-stance, while geckos ran upward (*a*) and sideways (*b*). TL, top left; TR, top right; BL, bottom left; BR, bottom right; TH, top hind; TF, top front; BH, bottom hind; BF, bottom front. Dashed lines indicate the orientation, whereas the thick solid bars show the magnitude of toe contact area. Percentages represent how frequently each toe made contact. *N* is the number of individuals, and *n* is the number of total trials. (*c*) Comparison of the toe configuration at each foot (violet: right rotated upward; orange: sideways) created by rotating figure 3*a* clockwise and overlaying it on figure 3*b*. Black arrows show the adjustment at multiple toes. See [26], movie S1. (*d*) Comparison of equivalent vertical contact area in sideways and upward running. Statistics are shown in [26], table S5.

(figure 4; [26], movie S2), behaving similarly to the situation when they were pulled for maximum shear force measurement [14]. The sliding speeds could be $0.42 \pm 0.26$ m s$^{-1}$ (figure 4) in less than 2 ms after the sliding occurred. All toes at the top feet reoriented more vertically in 15–25 ms with the middle ones converging to about 25° from the upward direction. We determined the resultant contact area at the top feet of sideways running geckos (figure 3*b*1,*b*2) and found that the vector of resultant contact area that represents the shear force at both top feet was also near 25° from the vertical direction. This confirmed that the upward aligned, clustered toes increased the resultant force, thus allowing animals to avoid falling during a severe surface perturbation. Given the brief duration of the response, we hypothesize that passive mechanical feedback [36] generates the adjustment.

### (ii) Slippery strips perturbation

Once any toe of a foot was able to adhere, geckos gained secure footholds and manoeuvred quickly sideways when travelling over slippery strips perpendicular to their motion with gaps of 10 mm and strip widths of 5 mm (figure 1*a*, *e*3). Kinematic analyses from high-speed videos indicated that the animals attached effectively and moved without measurable deceleration ([26], movie S2, $p = 0.37$). Using FTIR, we digitized the attachments of available toes at mid-stance (figure 5*b*; [26], figure S2) and compared them to geckos running sideways without slippery strips (figure 5*a*; see [26], table S6 for statistics).

We discovered considerable adjustments that included both their orientation and contact area at the available toes within corresponding feet, especially at those toes neighbouring the one losing contact (figure 5*b*; [26], figure S3 and

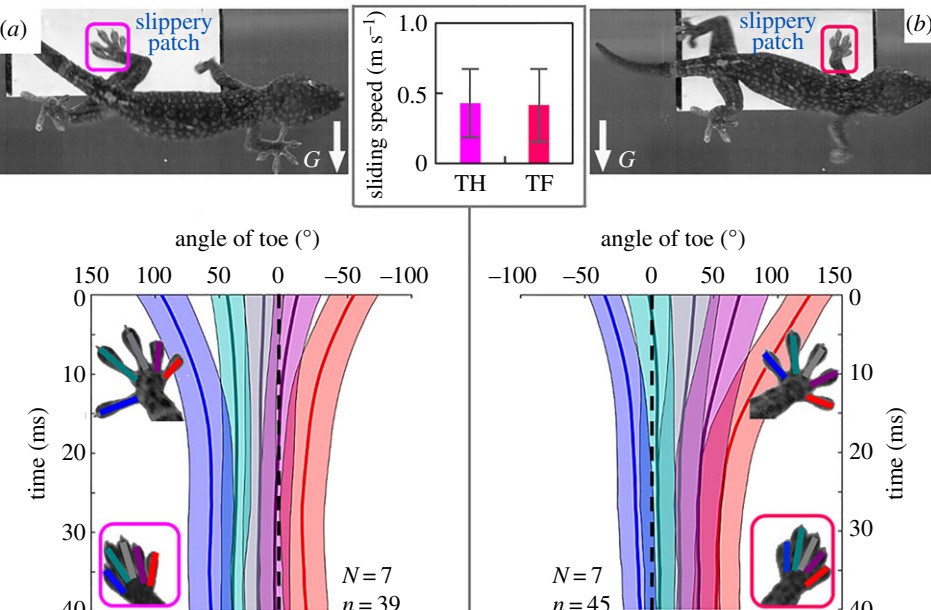

**Figure 4.** Toe angles change over time in response to a slippery patch during sideways running. (*a*) shows the top hind feet and (*b*) shows the top front feet. The inset shows the average sliding speed of top front (TF) and top hind (TH) feet in less than 2 ms after the sliding occurred ([26], movie S2). *N* is the number of individuals, and *n* is the number of trials. The squares in the top and bottom figures indicate the sliding feet. '*G*' and the white arrows indicate the gravity direction.

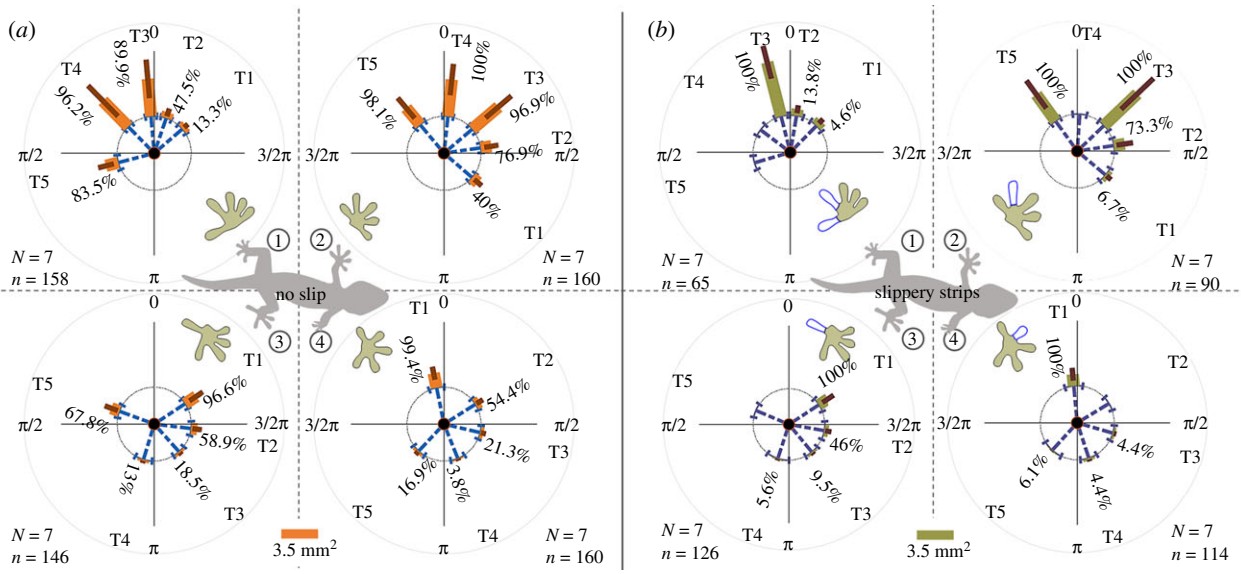

**Figure 5.** Comparison between toe deployments of geckos traversing wall without and with slippery strips. (*a,b*) Deployment of toes on different feet when there are no slips (*a*, orange bars) and slippery strips (*b*, tan bars). See [26], movie S2. Percentages represent the frequency of contact (See [26], figure S2). Dashed lines and solid lines indicate the average angles and contact area of digits, respectively. Feet: 1, top hind; 2, top front; 3, bottom hind; 4, bottom hind. The missing bars indicate the corresponding toes that lacked contact due to touching slippery strips. *N* is the number of individuals, and *n* is the number of trials. Statistics are shown in [26], table S6.

table S2). Geckos significantly increased the contact area of the third ($p < 0.001$) and fifth ($p < 0.001$) toes at top front feet to compensate for the contact loss of the fourth toe (figure 5*b*2; [26], figure S3*b*2). The top front feet increased the contact area of the second ($p < 0.001$) and fourth ($p < 0.001$) toes to compensate for the contact loss of the third toe ([26], figure S3*b*1) and further increased contact at the fourth ($p < 0.001$) toe if the fifth toe also lost contact ([26], figure S3*b*4). To compensate for the contact loss of the fourth and fifth toes at top hind feet, the effective contact ($p < 0.001$) and the orientation ($p < 0.001$) of the third toe

was adjusted (figure 5*b*1; [26], figure S3*a*4). If merely the fifth toes at top hind feet were inoperative, the third toe was adjusted in contact area ($p = 0.002$) and orientation ($p < 0.001$), but the fourth was not (contact, $p = 0.18$; orientation, $p = 0.70$; [26], figure S3*a*3). The lack of contact of the third ([26], figure S3*a*1) or fourth ([26], figure S3*a*2) toes was compensated by their neighbouring toes. Although the bottom feet did not typically contribute as much as the top feet in sideways running, geckos also adjusted the contribution of available toes to compensate for the loss in function of toes on slippery strips, and thereby secured

**Figure 6.** Feet attachment of geckos attained through toes while negotiating macro-scale rough substrates made with rods. (a) Typical foot attachment of geckos when running over vertically aligned rods perpendicular to the direction of motion ([26], movie S3). The red curves and yellow curves indicate the interface between bent toes and rods and that between the rolled toes and rods. 'G' and the white arrow indicate the gravity direction. Rod dimensions in mm. (b) The attachment capability of gecko feet increased with the increase of the rod diameter ([26], movie S4). The brown arrows indicate the direction of pulling. The attachment ability was calculated by dividing the maximum force of a foot on rods by its maximum force on the flat control surface. Significance level, $\star\star\star p \leq 0.01$, $\star\star 0.05 > p \geq 0.01$, $\star p \geq 0.05$.

effective footholds (figure 5b3,b4; [26], figure S3c,d). As shown by the percentages in figure 5 and [26], figure S2, geckos also adjusted the adhesion frequencies of these available toes to compensate for adhesion loss caused by the slippery strips.

Notably, adjustments were not achieved by simply increasing the contribution of all toes. The contact area of toes was sometimes reduced ([26], figure S3b2,b3) to produce the resultant adhesion of the corresponding feet. As indicated by the vector loops in [26], figure S3, geckos successfully compensated for the force loss caused by the contact loss of some toes by adjusting the alignment of their distributed toes ([26], table S4). Although toes and feet can sense force and respond to sensory information [37], we have insufficient evidence that neural feedback is used to adjust toes or future steps. The contact and force loss of some toes results in other toes bearing more force, and the altered load can mechanically deflect toe orientation if they are not parallel to the direction of load. To further clarify the role of passive versus active feedback control, electromyographic (EMG) recordings will be necessary in the future. Nonetheless, toe compensation clearly shows the critical advantages of distributed control among multiple adjustable structures.

## (d) Toes adapt to rough terrain

Animals must often negotiate terrain that is not flat and continuous, such as bark, discrete branches and uneven rocks [10]. The roughness of substrates could significantly alter the available area for animals to attach. When arthropods scurry on area-reduced substrates, some can use leg hairs or spines to effectively provide distributed mechanical feedback with the substrates [36]. For geckos, compliant lamellar structures can conform effectively to micro-rough surfaces [38,39] and provide greater opportunities for close contact with intermediately sized grooves [22].

Here, we challenged geckos with macroscopically rough, area-reduced terrain using sideways running over substrates

made with acrylic rods whose sizes (diameters 6.4–12.7 mm) are comparable to their toe lengths (figure 1a,e4; [26], movie S3). Geckos used a diversity of solutions to attain effective attachment on the area-reduced terrains by adjusting (bending and/or rotating) their digits (figure 6a; [26], figure S4). If there were no gaps, but unevenness, geckos always fit the convexities and concavities by bending and rotating soft toes (figure 6a1; [26], figure S4a,b). When we enlarged the gaps by decreasing the diameter of rods to 9.5 mm, toes had a higher probability of missing contact. The feet grabbed protrusions by bending toes to grasp one rod or by distributing toes across more than one rod (figure 6a2; [26], figure S4c,d). When there were 6.4 mm gaps between the rods, the toes behaved as they did on 9.5 mm rods, but could wrap the rods much more thoroughly (figure 6a3; [26], figure S4e,f). This grasping and surface conforming were also found for other feet ([26], figure S4a,b), indicating that geckos can actively attain reliable attachment.

Using the method of Gillies et al. [22], we quantified the attachment capability of gecko feet by measuring the shear force while pulling them across and along the rods used above (figure 1f; [26], movie S4). The results confirmed the potential of grasping rods with distributed toes. The maximum force on rods ranged from 3.36 to 12.59 N, with an average force of 9.92 N on the flat control. To best compare the attachment capability, we calculated the relative force by dividing the maximum force on rods of each individual with its maximum force on the flat control, as shown in figure 6b (see [26], table S7 for statistics). The relative force in the across-rod pulling increased from $60.1 \pm 9.3\%$ on 6.4 mm rods to $102.1 \pm 11.0\%$ on 12.7 mm rods ($p < 0.001$) before it remained unchanged on rods with larger diameters ($p = 0.48$) (figure 6b, red). By contrast, the relative force of the along-rod pulling kept increasing with the increase of rods size, eventually reaching 97.1% on 38 mm rods ($p < 0.001$; figure 6b, blue). Surprisingly, even on our smallest rod, gecko feet showed more than 50% attachment capability across and/or along the rods, being able to generate force

at least five times the bodyweight of the geckos (less than 100 g).

These results showed great adaptability of geckos feet on area-reduced and uneven substrates by controlling distributed toes to conform or grasp the substrate features. Arboreal animals with non-adhesive feet grasp tree branches to obtain and increase frictional contacts [40,41]. For geckos with adhesive pads, placing toes into the gaps and wrapping curved surfaces, more like human hands [42,43], increases the opportunities of intimate contact for more setae. Moreover, contact geometry at the peel zone of the seta and spatula become more favourable for both adhesion and friction as they are pulled at angles below 30° [44,45]. These discoveries suggest the possibility that adhesion on some rough surfaces could actually exceed those on smooth surfaces. Toes represent redundant foot extrusions that will not only increase the probability of intimately contacting more setae [25], but also increase the friction and adhesion. Multiple distributed extensions also offer the opportunity of attaching with mixed mechanisms that can include differential claws on mesopic rough terrain [12,33]. This kind of advantage was found to be indispensable for climbing robots which wish to extend their adaptability to rough surfaces [19].

## 4. Conclusion

A gecko's agile locomotion benefits from their unique adhesion by van der Waals attraction, but emerges as a result of the multi-level hierarchical arrangement of their locomotor appendages. Knowledge of how the setae adhesion is translated into foot adhesion using toes during acrobatic manoeuvres under varying conditions adds to our understanding of the hierarchy. Here, we provided evidence supporting our original hypotheses. Toes shared the load during steady-state locomotion and when responding to perturbations. Toes realigned when the load was altered due to gravity maintaining adequate force generation during both climbing and sideways running (figures 2 and 3). Toes

changed orientation and effective contact area to resist slippery patch and strip perturbations during sideways wall-running (figures 4 and 5). Compliant toes bent to match the rough terrain, even grasping the protrusions (figure 6). We conclude that multiple, soft toes demonstrate the important principles of multi-level conformability and redundancy. Gecko toes radiating from a foot show the effectiveness of distributed control afforded by multiple, adjustable compliant toes to increase manoeuvrability in demanding environments. Distributed control shows how biological adhesion can be deployed more effectively and offers design ideas for new robot feet, novel grippers and unique manipulators.

**Ethics.** The tests done at NUAA were approved by Jiangsu Association for Laboratory Animal Science and the Jiangsu Forestry Department (Approved File No. 2019-152). The tests conducted at UCB were approved by the Animal Care and Use Committee as mandated by the U.S. Animal Welfare Act and Public Health Service Policy (IACUC AUP-2017-03-9711). No animals were injured in any experiments.

**Data accessibility.** Electronic supplementary data are available from the Dryad Digital Repository: https://doi.org/10.6078/D1ZD6C [26].

**Authors' contributions.** R.J.F. proposed the research idea and designed the sideways wall-running experiments, Y.S. performed the tests. Z.D. designed, Y.S. and Z.W. performed the force-area measurement experiment. Y.S. processed the data. All authors analysed the data and discussed the results. Y.S., R.J.F. and Z.D. wrote the paper.

**Competing interests.** All authors declare that they have no competing interest.

**Funding.** This research was supported by UC Berkeley Institutional Funds to Centre for interdisciplinary Bio-inspiration in Education and Research (CiBER) and in part by the US Army Research Office under grant no. W911NF-17-1-0229 (R.J.F.), National Natural Science Foundation of China to Z.D. under grant no. 51435008 and scholarship from China Scholarship Council to Y.S.

**Acknowledgements.** We thank Xiaobo Lu, Jun Zhou and Camille Mercier for participating in this programme. Ben McInroe, Andrew Saintsing, Lawrence Wang and Ruby Ruopp provided advice on our experiments. We thank Kellar Autumn and Tom Libby for reading the manuscript.

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
