## [Reviewer comments · Proceedings of the Royal Society B: Biological Sciences]

Review History

RSPB-2020-0123.R0 (Original submission)

Review form: Reviewer 1

Recommendation

Accept with minor revision (please list in comments)

Scientific importance: Is the manuscript an original and important contribution to its field?

Good

General interest: Is the paper of sufficient general interest?

Good

Quality of the paper: Is the overall quality of the paper suitable?

Good

Is the length of the paper justified?

Yes

Should the paper be seen by a specialist statistical reviewer?

No

Do you have any concerns about statistical analyses in this paper? If so, please specify them explicitly in your report.

No

It is a condition of publication that authors make their supporting data, code and materials available - either as supplementary material or hosted in an external repository. Please rate, if applicable, the supporting data on the following criteria.

Is it accessible?

Yes

Is it clear?

Yes

Is it adequate?

Yes

Do you have any ethical concerns with this paper?

No

Comments to the Author

The present study investigated the role of geckos' toes in distributed control through sideways wall running experiments aiming at understanding of how the multiple, soft toes coordinate to facilitate gecko's acrobatic locomotion. This research had been well performed with careful experimental design and data analysis. The results are new and interesting that may provide a new perspective in developing gecko-inspired robotics and some other biomimetic devices. I therefore recommend to accept the paper with the condition to clarify the following comments.

1. There are too many direct quotes in the manuscript. In scientific writing, it is not appropriate to cite many origin paraphrases from others' papers. I would like the authors to rewrite these sentences in avoiding direct quotes.
2. Lines 61-72 of the introduction in page 3: This part of the text needs to be re-written to make these sentences clear and understandable.
3. Intuitively, the low friction patches or strips on the substrate lead to slipping of gecko toes, and as a result the distributed toes should be towards upward. So how to differentiate the passive and active contribution during this slipping distribution behavior of toes? The authors need to discuss it in sections "Slippery patch perturbation" and "Slippery strips perturbation" on pages 7 and 8.
4. Figure 1 b shows the X-axis has two arrows in opposite direction both marked "+x". Is it correct?
5. Conclusion part need rewritten. You should draw a conclusion from your study not cited others' statements.

Review form: Reviewer 2

Recommendation

Accept with minor revision (please list in comments)

Scientific importance: Is the manuscript an original and important contribution to its field?

Good

General interest: Is the paper of sufficient general interest?

Good

Quality of the paper: Is the overall quality of the paper suitable?

Acceptable

Is the length of the paper justified?

Yes

Should the paper be seen by a specialist statistical reviewer?

No

Do you have any concerns about statistical analyses in this paper? If so, please specify them explicitly in your report.

Yes

It is a condition of publication that authors make their supporting data, code and materials available - either as supplementary material or hosted in an external repository. Please rate, if applicable, the supporting data on the following criteria.

Is it accessible?

No

Is it clear?

No

Is it adequate?

No

Do you have any ethical concerns with this paper?

Yes

Comments to the Author

Summary

This study investigates how live geckos alter contact area, force, and toe orientation to run on vertical and side-ways substrates, and on substrates with altered conditions (slippery surfaces, large-scale roughness). The results of this study are interesting and relevant to the field, as noted by several direct quotations of recent review articles in this manuscript. This work is also relevant for engineering application, particularly robotic application of multiple contact points to improve control and adaptability in a variety of conditions. The authors could even add a line to the conclusion to highlight this connection. While this study provides several new findings, it is at times hard to follow and missing some detail. I make the following suggestions to improve the text.

Comments

Decide if sideways wall-running has a hyphen or not and stay consistent.

Author-supplied Statements - This should have ethical considerations for vertebrate animals. Please change to yes and add protocol numbers (IACUC and international equivalent) here and to the methods.

Page 2, Line 46 - the line "More digits ... toe span." is a little unclear/ambiguous. What is a toe span, what is the top half? How is this different from the line before where the third toe is aligned with the gravity vector? The following lines are a little confusing as well, what did the study find and how are the results different (i.e., what is the direct contrast between predictions and results)?

Page 4, Line 95 - adequate forces to support their body weight? maintain constant velocity? something else?

Page 4, Line 102 - "Feed divided into toes.." is a little out of place, move it down?

Page 4, Line 108 - I am a little confused about what these statistical analyses relate to. In Fig. 2 there are 6 total regressions, which are these? Also, the second p-value is nearly right on the border, which could be interesting but I am not sure what this analysis is.

Page 5, line 119 - do you mean on horizontally oriented acrylic?

Page 5, line 126 - where in the SI is this result?

Page 5, line 129 - I believe from this point on the statistical information (F value, degrees of freedom, R^2 if applicable) are left off and only P-values are given. This makes it hard to tell what was analyzed and how. Please add appropriate statistical information to the results throughout.

Page 5, line 131 - Should this be Fig 3 c1&2? Also, I am a little confused by Fig. 3c. What toes are 1-4? Visually I am trying to overlay a and b to make c. I can't seem to match what I expect from a and b to the figure in c. For instance, it looks like c is a sideways gecko (i.e., top two feet have high contact area), so the purple bars should be a vertically running gecko shifted to match a sideways running gecko, but if I shift the vertical results sideways, they don't match the purple bars in c. I am not sure if this is how I should be interpreting this figure. I think fundamentally it is interesting to show how area and orientation are altered on a sideways run compared to a vertical. Fig. 3c needs to show that.

Page 5, line 146 - 148 - Fig. 3 d is feet not toes, right?

Page 6, line 176 - reference 9 is in an odd place to read. Also, how is it being used?

Page 7, line 190 - this is not a neighboring toe, right? (according to first line in this paragraph, you always found area and orientation change in neighbors).

Page 7, line 187-198 - this paragraph is confusing. What I take from it is that contact area (force) and orientation vary among toes when contact is lost. So 2nd and 4th compensate for 3rd toe, and 2nd and 4th compensate for 5th. These have stats associated with them, though hard to understand because only p-values are reported (as noted above). The next line (191) states that area and orientation offset reduction of 4th or 5th, but now you do not state by what toes, and isn't this comparison already made above? The next line includes the 3rd, 4th, 5th, toes again, I am not sure why/what this comparison is except is this now just in the hind foot? What were the others listed previously? Finally, line 194 says that geckos can hold a step of the hind foot (top) with single toes, which toes? Can they not hold with fore foot too? Please clarify this paragraph.

Page 7, line 204 - this last line is a little confusing. I think you mean to compare passive and active. Perhaps put in parentheses which part is passive and which is active.

Page 8, line 220 - change warp to wrap

Page 8, line 222 and 223 - change strips to rods and stay consistent. Strips makes me think of the teflon slip strips

Page 8, line 241 - remove "the" in "the tree branches"

Page 9, line 267 - what is the "first experiment"? It seems you have several measures where you

need to compare the same individual, which cannot be done among geckos tested in China vs. US, so please clarify which were used in which experiments. In lines 286-291 you discuss control of the same individual (with/ without slip) so this must have been done with one set of geckos (i.e., same individuals in one country). Also, in this example, how was the control used? For instance, with the rods you calculate a percent for each animal. In this one I think you just compare? The figure shows variable n for each, so I am guessing there is no direct comparison within individuals. Please clarify.

Page 9, line 281 - do you mean horizontal plexiglass track?

Page 10, line 314 - is "resultant contacts" area? I am guessing this goes with Figure 2. Please clarify which stats test goes with which comparison and use the same terms (e.g., Foot Contact Area).

Page 11, line 324 - Song's name should not be in all capital letters

Figure 1. What units are D, d, and w in? Please add to figure or legend.

Figure 2. Please make the separation between a-f and g-i a little larger, I was not sure where the "Foot Contact A" in c and f was meant to be (looks like it could go with i at first glance before noticing the faint separation line).

Figure 2. The line "For the top front feet,..." is out of sequence? This does not relate to g,h,i, it is for the previous parts of the figure, correct? Also, are the dashed lines in g-i the x and the solid are y? Hard to tell and they are not distinguished.

Figure 3. Where are the percentages?

Figure 4. What is the initial sliding speed? When is this speed taken (i.e., it is not zero)?

Figure 5. A bit more detail about the bars and dashed lines like the above figure would be helpful in case a reader is only looking at this figure or as a reminder of what they mean.

Figure 6. Change "capability" to "attachment capability" in the ledged to stay consistent and clear. Again, move up any text that relates to a so that it is all in one place. The line "In Fig. a, the red lines..." should be moved up, then introduce b and all of the information associated with that panel. Clarify the pink vs. blue colors.

Decision letter (RSPB-2020-0123.R0)

29-Feb-2020

Dear Professor Full:

Your manuscript has now been peer reviewed and the reviews have been assessed by an Associate Editor. The reviewers' comments (not including confidential comments to the Editor) and the comments from the Associate Editor are included at the end of this email for your reference. As you will see, the reviewers and the Editors have raised some concerns with your manuscript and we would like to invite you to revise your manuscript to address them.

We do not allow multiple rounds of revision so we urge you to make every effort to fully address all of the comments at this stage. If deemed necessary by the Associate Editor, your manuscript will be sent back to one or more of the original reviewers for assessment. If the original reviewers

are not available we may invite new reviewers. Please note that we cannot guarantee eventual acceptance of your manuscript at this stage.

Research ethics:

Use of animals and field studies:

Please submit a copy of your revised paper within three weeks. If we do not hear from you within this time your manuscript will be rejected. If you are unable to meet this deadline please let us know as soon as possible, as we may be able to grant a short extension.

Best wishes,
Dr Sasha Dall
mailto:proceedingsb@royalsociety.org

Associate Editor
Board Member: 1
Comments to Author:

Both reviewers agree that this manuscript is scientifically important and has broad general interest. They have various suggestions for improving it, mainly related to the clarity of the text and figures. Please read and respond to these suggestions. Two important concerns were identified that relate to our publication policies. First, the raw data are not accessible, which is necessary per our open data policy unless there is a compelling reason to grant an exception. Second, while the institutional protocols are correctly listed in the Methods section, the author supplied statement on Ethics does not include the permits necessary for experiments on vertebrate animals.

Reviewer(s)' Comments to Author:

Referee: 1

Comments to the Author(s)

The present study investigated the role of geckos' toes in distributed control through sideways wall running experiments aiming at understanding of how the multiple, soft toes coordinate to facilitate gecko's acrobatic locomotion. This research had been well performed with careful experimental design and data analysis. The results are new and interesting that may provide a new perspective in developing gecko-inspired robotics and some other biomimetic devices. I therefore recommend to accept the paper with the condition to clarify the following comments.

1. There are too many direct quotes in the manuscript. In scientific writing, it is not appropriate to cite many origin paraphrases from others' papers. I would like the authors to rewrite these sentences in avoiding direct quotes.
2. Lines 61-72 of the introduction in page 3: This part of the text needs to be re-written to make these sentences clear and understandable.
3. Intuitively, the low friction patches or strips on the substrate lead to slipping of gecko toes, and as a result the distributed toes should be towards upward. So how to differentiate the passive and

active contribution during this slipping distribution behavior of toes? The authors need to discuss it in sections "Slippery patch perturbation" and "Slippery strips perturbation" on pages 7 and 8.

4. Figure 1 b shows the X-axis has two arrows in opposite direction both marked "+x". Is it correct?

5. Conclusion part need rewritten. You should draw a conclusion from your study not cited others' statements.

Referee: 2

Comments to the Author(s)

Summary

This study investigates how live geckos alter contact area, force, and toe orientation to run on vertical and side-ways substrates, and on substrates with altered conditions (slippery surfaces, large-scale roughness). The results of this study are interesting and relevant to the field, as noted by several direct quotations of recent review articles in this manuscript. This work is also relevant for engineering application, particularly robotic application of multiple contact points to improve control and adaptability in a variety of conditions. The authors could even add a line to the conclusion to highlight this connection. While this study provides several new findings, it is at times hard to follow and missing some detail. I make the following suggestions to improve the text.

Comments

Decide if sideways wall-running has a hyphen or not and stay consistent.

Author-supplied Statements - This should have ethical considerations for vertebrate animals. Please change to yes and add protocol numbers (IACUC and international equivalent) here and to the methods.

Page 2, Line 46 - the line "More digits ... toe span." is a little unclear/ambiguous. What is a toe span, what is the top half? How is this different from the line before where the third toe is aligned with the gravity vector? The following lines are a little confusing as well, what did the study find and how are the results different (i.e., what is the direct contrast between predictions and results)?

Page 4, Line 95 - adequate forces to support their body weight? maintain constant velocity? something else?

Page 4, Line 102 - "Feed divided into toes.." is a little out of place, move it down?

Page 4, Line 108 - I am a little confused about what these statistical analyses relate to. In Fig. 2 there are 6 total regressions, which are these? Also, the second p-value is nearly right on the border, which could be interesting but I am not sure what this analysis is.

Page 5, line 119 - do you mean on horizontally oriented acrylic?

Page 5, line 126 - where in the SI is this result?

Page 5, line 129 - I believe from this point on the statistical information (F value, degrees of freedom, R^2 if applicable) are left off and only P-values are given. This makes it hard to tell what was analyzed and how. Please add appropriate statistical information to the results throughout.

Page 5, line 131 - Should this be Fig 3 c1&2? Also, I am a little confused by Fig. 3c. What toes are 1-4? Visually I am trying to overlay a and b to make c. I can't seem to match what I expect from a and b to the figure in c. For instance, it looks like c is a sideways gecko (i.e., top two feet have

high contact area), so the purple bars should be a vertically running gecko shifted to match a sideways running gecko, but if I shift the vertical results sideways, they don't match the purple bars in c. I am not sure if this is how I should be interpreting this figure. I think fundamentally it is interesting to show how area and orientation are altered on a sideways run compared to a vertical. Fig. 3c needs to show that.

Page 5, line 146 - 148 - Fig. 3 d is feet not toes, right?

Page 6, line 176 - reference 9 is in an odd place to read. Also, how is it being used?

Page 7, line 190 - this is not a neighboring toe, right? (according to first line in this paragraph, you always found area and orientation change in neighbors).

Page 7, line 187-198 - this paragraph is confusing. What I take from it is that contact area (force) and orientation vary among toes when contact is lost. So 2nd and 4th compensate for 3rd toe, and 2nd and 4th compensate for 5th. These have stats associated with them, though hard to understand because only p-values are reported (as noted above). The next line (191) states that area and orientation offset reduction of 4th or 5th, but now you do not state by what toes, and isn't this comparison already made above? The next line includes the 3rd, 4th, 5th, toes again, I am not sure why/what this comparison is except is this now just in the hind foot? What were the others listed previously? Finally, line 194 says that geckos can hold a step of the hind foot (top) with single toes, which toes? Can they not hold with fore foot too? Please clarify this paragraph.

Page 7, line 204 - this last line is a little confusing. I think you mean to compare passive and active. Perhaps put in parentheses which part is passive and which is active.

Page 8, line 220 - change warp to wrap

Page 8, line 222 and 223 - change strips to rods and stay consistent. Strips makes me think of the teflon slip strips

Page 8, line 241 - remove "the" in "the tree branches"

Page 9, line 267 - what is the "first experiment"? It seems you have several measures where you need to compare the same individual, which cannot be done among geckos tested in China vs. US, so please clarify which were used in which experiments. In lines 286-291 you discuss control of the same individual (with/without slip) so this must have been done with one set of geckos (i.e., same individuals in one country). Also, in this example, how was the control used? For instance, with the rods you calculate a percent for each animal. In this one I think you just compare? The figure shows variable n for each, so I am guessing there is no direct comparison within individuals. Please clarify.

Page 9, line 281 - do you mean horizontal plexiglass track?

Page 10, line 314 - is "resultant contacts" area? I am guessing this goes with Figure 2. Please clarify which stats test goes with which comparison and use the same terms (e.g., Foot Contact Area).

Page 11, line 324 - Song's name should not be in all capital letters

Figure 1. What units are D, d, and w in? Please add to figure or legend.

Figure 2. Please make the separation between a-f and g-i a little larger, I was not sure where the "Foot Contact A" in c and f was meant to be (looks like it could go with i at first glance before noticing the faint separation line).

Figure 2. The line "For the top front feet,..." is out of sequence? This does not relate to g,h,i, it is for

the previous parts of the figure, correct? Also, are the dashed lines in g-i the x and the solid are y? Hard to tell and they are not distinguished.

Figure 3. Where are the percentages?

Figure 4. What is the initial sliding speed? When is this speed taken (i.e., it is not zero)?

Figure 5. A bit more detail about the bars and dashed lines like the above figure would be helpful in case a reader is only looking at this figure or as a reminder of what they mean.

Figure 6. Change "capability" to "attachment capability" in the ledged to stay consistent and clear. Again, move up any text that relates to a so that it is all in one place. The line "In Fig. a, the red lines..." should be moved up, then introduce b and all of the information associated with that panel. Clarify the pink vs. blue colors.

Author's Response to Decision Letter for (RSPB-2020-0123.R0)

See Appendix A.

Decision letter (RSPB-2020-0123.R1)

03-Apr-2020

Dear Professor Full

I am pleased to inform you that your manuscript entitled "Role of multiple, adjustable toes in distributed control shown by sideways wall-running in geckos" has been accepted for publication in Proceedings B.

Open Access

You are invited to opt for Open Access, making your freely available to all as soon as it is ready for publication under a CCBY licence. Our article processing charge for Open Access is £1700. Corresponding authors from member institutions (<http://royalsocietypublishing.org/site/librarians/allmembers.xhtml>) receive a 25% discount to these charges. For more information please visit <http://royalsocietypublishing.org/open-access>.

Paper charges

Sincerely,

Dr Sasha Dall

Appendix A

Dear editors and referees,

Thank you for your comments that helped to improve the quality and readability of our manuscript significantly.

We responded to all the comments one-by-one in this file and made very careful modifications in the main text, figures, and supporting materials as requested.

Associate Editor

Board Member: 1

Comments to Author:

Both reviewers agree that this manuscript is scientifically important and has broad general interest. They have various suggestions for improving it, mainly related to the clarity of the text and figures. Please read and respond to these suggestions. Two important concerns were identified that relate to our publication policies. First, the raw data are not accessible, which is necessary per our open data policy unless there is a compelling reason to grant an exception. Second, while the institutional protocols are correctly listed in the Methods section, the author supplied statement on Ethics does not include the permits necessary for experiments on vertebrate animals.

Answer: We deposited our data at

<https://doi.org/10.6078/D1ZD6C>

We have checked the accessibility.

Our manuscript involves separate experiments done at NUAA and UCB. Tests done at NUAA were approved by Jiangsu Association for Laboratory Animal Science and Jiangsu Forestry Department (Approve File No. 2019-152), and those conducted at UCB were approved by the Animal Care and Use Committee as mandated by the US Animal Welfare Act and Public Health Service Policy (IACUC AUP-2017-03-9711).

We updated our statement on Ethics.

Reviewer(s)' Comments to Author:

Referee: 1

Comments to the Author(s)

The present study investigated the role of geckos' toes in distributed control through sideways wall running experiments aiming at understanding of how the multiple, soft toes

coordinate to facilitate gecko's acrobatic locomotion. This research had been well performed with careful experimental design and data analysis. The results are new and interesting that may provide a new perspective in developing gecko-inspired robotics and some other biomimetic devices. I therefore recommend to accept the paper with the condition to clarify the following comments.

1. There are too many direct quotes in the manuscript. In scientific writing, it is not appropriate to cite many origin paraphrases from others' papers. I would like the authors to rewrite these sentences in avoiding direct quotes.

Response: Thank you for your suggestion. We have removed the direct quotes in the revised manuscript and rewrote the text.

2. Lines 61-72 of the introduction in page 3: This part of the text needs to be rewritten to make these sentences clear and understandable.

Response: As requested, we have rewritten the section to make it clearer by better defining our points.

3. Intuitively, the low friction patches or strips on the substrate lead to slipping of gecko toes, and as a result the distributed toes should be towards upward. So how to differentiate the passive and active contribution during this slipping distribution behavior of toes? The authors need to discuss it in sections "Slippery patch perturbation" and "Slippery strips perturbation" on pages 7 and 8.

Response: Added discussion as requested.

For slippery patches, we now note that given the brief duration of the response (<25 ms), we hypothesize that passive mechanical feedback generates the adjustment.

For slippery strips, we explain that we have insufficient evidence pointing to neural feedback being used to adjust toes or future steps, even though toes and feet can sense force and respond to sensory information. The contact and force loss of some toes results in other toes bearing more force, and the altered load can mechanically deflect toe orientation if they are not parallel to the direction of load. To further clarify the role of passive versus active feedback control, electromyographic (EMG) recordings will be necessary in the future.

4. Figure 1 b shows the X-axis has two arrows in opposite direction both marked “+x”. Is it correct?

Response: We have revised this axis to make it easier to understand by removing the +/- . We define the lateral *x-direction* for each foot relative to the body. Positive lateral wall-reaction forces (+x; red) correspond to forces where feet pull inward toward the trunk, whereas negative lateral wall-reaction forces (–x; red) correspond to forces where feet push away from the trunk.

5. Conclusion part need rewritten. You should draw a conclusion from your study not cited others' statements.

Response: As requested, we have significantly modified and strengthened our conclusion. We have gone back to our hypotheses, elaborated on them, and then added the principles that they demonstrate.

Referee: 2

Comments to the Author(s)

Summary

This study investigates how live geckos alter contact area, force, and toe orientation to run on vertical and sideways substrates, and on substrates with altered conditions (slippery surfaces, large-scale roughness). The results of this study are interesting and relevant to the field, as noted by several direct quotations of recent review articles in this manuscript. This work is also relevant for engineering application, particularly robotic application of multiple contact points to improve control and adaptability in a variety of conditions. The authors could even add a line to the conclusion to highlight this connection. While this study provides several new findings, it is at times hard to follow and missing some detail. I make the following suggestions to improve the text.

Response: Thanks for your advice. We have rewritten the conclusion to better articulate the principles we learned and highlight the bio-inspiration.

Comments

1. Decide if sideways wall-running has a hyphen or not and stay consistent.

Response: Thanks. We decided to use ‘wall-running’ throughout.

2. Author-supplied Statements - This should have ethical considerations for vertebrate animals. Please change to yes and add protocol numbers (IACUC and international equivalent) here and to the methods.

Response: Added. We changed to yes and have updated the statement on ethics. We have a protocol (IACUC_AUP-2017-03-9711) for the experiments at UCB and an approved file (Approved File No. 2019-152) for the experiments at NUAA.

3. Page 2, Line 46 - the line "More digits ... toe span." is a little unclear/ambiguous. What is a toe span, what is the top half? How is this different from the line before where the third toe is aligned with the gravity vector? The following lines are a little confusing as well, what did the study find and how are the results different (i.e., what is the direct contrast between predictions and results)?

Response:

We have reworded the findings by Russell. Here, we used to span to mean the fan like toes at a foot. The first and the fifth toes at this foot are the edges of this fan.

Russell and Oetelaar predicted that "Digits will be preferentially clustered together (in terms of their mean compass direction) in the 'northern' half of the circle (Fig. 1), because only digits with their long axes directed above the horizontal (East-West) meridian will be able to be gravitationally loaded." The circle was centring at mid-point of the trunk and circumscribing the trunk.

According to their experiment, different station keeping shows different toe orientations. The third toes at top feet were not always along the vertical direction and toes at bottom feet were not always clustered pointing upward, as shown in their Fig. 4.

Such variation confirmed that the deployment of toes during station keeping is only related to attaining sufficient foot forces and used only a small subset of the capacity. We have rewritten this paragraph to make it clearer.

4. Page 4, Line 95 - adequate forces to support their body weight? maintain constant velocity? something else?

Response: Clarified. Balancing gravity is most critical. After that, geckos can move their bodies forward and do other manoeuvres.

We modified the sentence as '...thereby produce adequate forces to balance gravity and locomote.

5. Page 4, Line 102 - "Feet divided into toes.." is a little out of place, move it down?

Response: We have moved this sentence to the next paragraph.

6. Page 4, Line 108 - I am a little confused about what these statistical analyses relate to. In Fig. 2 there are 6 total regressions, which are these? Also, the second p-value is nearly right on the border, which could be interesting but I am not sure what this analysis is.

Response: Here, we used a general linear model (GLM) and linear regression (LR) to determine the relationship between the force and contact area and compared the linear relationships found at three different phases in the stance (3 sets of regressions, one for each phase g-h, and two regressions for each phase representing x and y force directions for a total of six regressions)

For the linear regression, we have added the information in the electronic supplemental material results and have better explained the regressions in the caption of Fig. 2.

$F_{(2,630)}=2.83$, $P=0.06$ indicated a comparison among the linear relationship at the three phases we selected using a GLM analysis. Though the P-value was near 0.05, we contend that such a difference would be small and likely unimportant given such a strong correspondence between force and contact area.

7. Page 5, line 119 - do you mean on horizontally oriented acrylic?

Response: Yes. We have rewritten the sentence to say, "While running sideways along an acrylic wall..."

8. Page 5, line 126 - where in the SI is this result?

Response: Due to the word limit, we did not describe the deployment of each toe in sideway wall-running and the control (upward climbing) in the main text. We described Fig. 3a&b in the esm results in detail (esm information, Page 4, Titled 'Toes realign when relative load orientation is altered').

9. Page 5, line 129 - I believe from this point on the statistical information (F value, degrees of freedom, R^2 if applicable) are left off and only P-values are given. This makes it hard to tell what was analysed and how. Please add appropriate statistical information to the results throughout.

Response: Thanks for your advice. Added Tables S4-7 in the esm for analysis of all data in figures, so as to not overwhelm the Results section with these values. We used ANOVA for the comparisons of the toe angles and toe contact area between different conditions. We

have added the F -value, $d.f.s$. The statistics results are now presented in the format as follows ($F_{(1,205)}=60.135, P<0.001$).

10. Page 5, line 131 - Should this be Fig 3 c1&2? Also, I am a little confused by Fig. 3c. What toes are 1-4? Visually I am trying to overlay a and b to make c. I can't seem to match what I expect from a and b to the figure in c. For instance, it looks like c is a sideways gecko (i.e., top two feet have high contact area), so the purple bars should be a vertically running gecko shifted to match a sideways running gecko, but if I shift the vertical results sideways, they don't match the purple bars in c. I am not sure if this is how I should be interpreting this figure. I think fundamentally it is interesting to show how area and orientation are altered on a sideways run compared to a vertical. Fig. 3c needs to show that.

Response: We have modified Fig. 3 and its caption so that the comparison is clearer. To make Fig. 3c, we rotated Fig. 3a clockwise and then overlaid it on Fig. 3b, as shown in the figure below.

To better compare the toe deployment of each foot in sideways wall-running (orange), a rotated upward climbing (violet) allowed us to show the toe realignment more directly.

11. Page 5, line 146 - 148 - Fig. 3 d is feet not toes, right?

Response: Yes, corrected. Here Fig. 3d should be Fig. 3c.

12. Page 6, line 176 - reference 9 is in an odd place to read. Also, how is it being used?

Response: We rewrote the sentence to make it clearer. In Reference 9, now 16 (Wang Z, Wang J, Ji A, Zhang Y, Dai Z. 2011 Behavior and dynamics of gecko's locomotion: The effects of moving directions on a vertical surface. Chinese Sci. Bull. 56, 573–583.), Wang *et al.* reported the force at the feet of sideways running geckos in their Fig. 4. We tried to alert readers that foot data were present and consistent with our toe measurements.

13. Page 7, line 190 - this is not a neighboring toe, right? (according to first line in this paragraph, you always found area and orientation change in neighbors).

Response: Correct, we have rewritten this paragraph. Actually, the area and orientation change occur at the toes that remain available. Toes neighbouring the one losing contact adjust their alignment most frequently.

14. (i) Page 7, line 187-198 - this paragraph is confusing. What I take from it is that contact area (force) and orientation vary among toes when contact is lost. So 2nd and 4th compensate for 3rd toe, and 2nd and 4th compensate for 5th. These have stats associated with them, though hard to understand because only p-values are reported (as noted above).

(ii) The next line (191) states that area and orientation offset reduction of 4th or 5th, but now you do not state by what toes, and isn't this comparison already made above?

(iii) The next line includes the 3rd, 4th, 5th, toes again, I am not sure why/what this comparison is except. Is this now just in the hind foot? What were the others listed previously?

(iv) Finally, line 194 says that geckos can hold a step of the hind foot (top) with single toes, which toes? Can they not hold with fore foot too? Please clarify this paragraph.

Response: Clarified as requested. To make the description clearer, we added an example comparison in Fig.5 and rewrote the whole paragraph.

(i) We have added F-values and degrees of freedom for statistical analysis in the Tables S4-7 of the revised esm.

(ii) We have revised this section to make the comparison clearer.

(iii) Yes, this sentence was describing the top hindfoot. We expanded the description here to improve the readability in the revised section.

(iv) In our experiment, we found the top hind feet can hold using the third toes. The top front feet can also hold the attachment solely *via* the fourth toes. We have revised the section accordingly.

15. Page 7, line 204 - this last line is a little confusing. I think you mean to compare passive and active. Perhaps put in parentheses which part is passive and which is active.

Response: We have rewritten this paragraph to make our statement more precise.

16. Page 8, line 220 - change warp to wrap

Response: Thanks. Corrected.

17. Page 8, line 222 and 223 - change strips to rods and stay consistent. Strips makes me think of the teflon slip strips

Response: Modified as requested.

18. Page 8, line 241 - remove "the" in "the tree branches"

Response: Deleted.

19. Page 9, line 267 - what is the "first experiment"? It seems you have several measures where you need to compare the same individual, which cannot be done among geckos tested in China vs. US, so please clarify which were used in which experiments. In lines 286-291 you discuss control of the same individual (with/without slip) so this must have been done with one set of geckos (i.e., same individuals in one country). Also, in this example, how was the control used? For instance, with the rods you calculate a percent for each animal. In this one I think you just compare? The figure shows variable n for each, so I am guessing there is no direct comparison within individuals. Please clarify.

Response: To clarify, we used 14 geckos in total. Seven of them participated in experiments in China (Fig.2), and the other seven participated in all experiments in the US (Fig. 3-6).

(1) While studying the deployment of toes in sideways wall-running (Fig.3, USA), we used the upward climbing of the same individual as control.

(2) In the slippery strips experiment (Fig.5, USA), we used the sideways wall-running without slippery strips of the same individuals (i.e., the results found in the above experiment, Fig.3) as control.

(3) In the last experiment on rods (Fig.6, USA), we used a flat acrylic sheet as the control surface. We calculated the attachment capability by dividing the maximum force of a foot on rods with its own maximum force on the flat control surface.

Due to the word limits, we have clarified these in the Methods section and added detailed information in the esm methods.

20. Page 9, line 281 - do you mean horizontal plexiglass track?

Response: Revised. We now say, "we built another Plexiglas wall with a track that allowed sideways running..."

21. Page 10, line 314 - is "resultant contacts" area? I am guessing this goes with Figure 2. Please clarify which stats test goes with which comparison and use the same terms (e.g., Foot Contact Area).

Response: Corrected to read "resultant foot contact area." Yes, with Fig. 2.

22. Page 11, line 324 - Song's name should not be in all capital letters

Response: Corrected.

23. Figure 1. What units are D, d, and w in? Please add to figure or legend.

Response: Millimetres added to legend.

23. Figure 2. Please make the separation between a-f and g-i a little larger, I was not sure where the "Foot Contact A" in c and f was meant to be (looks like it could go with i at first glance before noticing the faint separation line).

Response: We modified the figure as requested. First, we made the distance between the left and right part wider. Second, we divided the subfigures with a solid line.

24. Figure 2. The line "For the top front feet,..." is out of sequence? This does not relate to g,h,i, it is for the previous parts of the figure, correct? Also, are the dashed lines in g-i the x and the solid are y? Hard to tell and they are not distinguished.

Response: Corrected with revised caption and figure. We now use thicker lines to make them clearer.

25. Figure 3. Where are the percentages?

Response: As requested, we modified the figure by adding percentages.

26. Figure 4. What is the initial sliding speed? When is this speed taken (i.e., it is not zero)?

Response: At the instant when sliding occurs, the speed is zero. The speed here was calculated by dividing the sliding distance in the first frame after sliding with the frame time

(1/600 s). We have clarified this in the caption and changed the relative text as 'the sliding speed could reach 0.42 ± 0.26 m/s less than 2ms s after the sliding occurred.

27. Figure 5. A bit more detail about the bars and dashed lines like the above figure would be helpful in case a reader is only looking at this figure or as a reminder of what they mean.

Response: We have modified the legend as requested.

28. Figure 6. Change "capability" to "attachment capability" in the ledged to stay consistent and clear. Again, move up any text that relates to a so that it is all in one place. The line "In Fig. a, the red lines..." should be moved up, then introduce b and all of the information associated with that panel. Clarify the pink vs. blue colours.

Response: Figure and caption modified as requested. We also changed the colours to better distinguish the result.